# Diagnostic Splenectomy: Characteristics, Pre-Operative Investigations, and Identified Pathologies for 20 Patients

**DOI:** 10.3390/jcm10071519

**Published:** 2021-04-06

**Authors:** Jean Maillot, Jean-Valère Malfuson, Thierry Lazure, Stéphane Benoist, Anne Cremades, Emmanuel Hornez, Florent L. Besson, Nicolas Noël, Olivier Lambotte

**Affiliations:** 1Service de Médecine Interne et Immunologie Clinique, APHP, Hôpitaux Universitaires Paris Saclay, Hôpital Bicêtre, 94720 Le Kremlin-Bicêtre, France; jeanmaillot@hotmail.fr (J.M.); nicolas.noel@aphp.fr (N.N.); 2Service d’Hématologie Clinique, Hôpital d’Instruction des Armées Percy, 92140 Clamart, France; jvmalf@free.fr; 3Service d’Anatomo-Pathologie, APHP, Hôpitaux Universitaires Paris Saclay, Hôpital Bicêtre, 94720 Le Kremlin-Bicêtre, France; thierry.lazure@aphp.fr; 4Service de Chirurgie Digestive, APHP, Hôpitaux Universitaires Paris Saclay, Hôpital Bicêtre, 94720 Le Kremlin-Bicêtre, France; stephane.benoist@aphp.fr; 5Service d’Anatomo-Pathologie, Hôpital d’Instruction des Armées Bégin, 94160 Saint Mandé, France; anne.stephan@intradef.gouv.fr; 6Service de Chirurgie Digestive, Hôpital d’Instruction des Armées Percy, 92140 Clamart, France; emmanuel.hornez@intradef.gouv.fr; 7Service de Médecine Nucléaire, APHP, Hôpitaux Universitaires Paris Saclay, Hôpital Bicêtre, 94720 Le Kremlin-Bicêtre, France; florent.besson@aphp.fr; 8Faculté de Médecine, Université Paris-Saclay, 94720 Le Kremlin-Bicêtre, France; 9CEA, Center for Immunology of Viral, Auto-Immune, Haematological and Bacterial Diseases (ID-MIT/IMVA-HB), Université Paris-Saclay, Inserm, 94720 Kremlin-Bicêtre, France

**Keywords:** diagnostic splenectomy, diagnostic investigations, malignant hematologic disorders, lymphoproliferative disease, FDG PET/CT, bone-marrow biopsy

## Abstract

Splenectomy is indicated in cases of trauma to the spleen or hematological and immunological diseases (hereditary spherocytosis, autoimmune cytopenia). Less frequently, splenectomy is performed for diagnostic purposes to complement unsuccessful prior etiological investigations. The splenectomy remains a surgery at risk of complications and should be considered as a last-resort procedure to make the diagnosis and to be able to treat patients. We studied the medical files of 142 patients who underwent a splenectomy for any reason over a 10-year period and identified 20 diagnostic splenectomies. Diagnostic splenectomies were mainly performed to explore unexplained splenomegaly for 13 patients and fever of unknown origin for 10. The other patients had surgery for other causes (cytopenia, abdominal symptoms, suspicion of relapsing malignant hemopathies). Splenectomy contributed to the final diagnosis in 19 of 20 cases, corresponding mostly to lymphoid hemopathies (14/20). The most frequent disease was diffuse large B-cell lymphoma (8/20). Splenectomy did not reveal any infectious disease. The most relevant pre-operative procedures to aid the diagnosis were ^18^F-fluorodeoxyglucose positron emission tomography/computed tomography (FDG PET/CT) and immuno-hematological examinations. Diagnostic splenectomy is useful and necessary in certain difficult diagnostic situations. Highlights: Diagnostic splenectomy is still useful in 2020 to diagnose unexplained splenomegaly or fever of unknown origin. Lymphoma was the most common final diagnosis. FDG PET/CT was the most useful tool to aid in the diagnosis.

## 1. Introduction

The spleen is the largest lymphatic organ of the body. It has three main functions [1] blood cell storage, blood filtration, and immunological processes. Indications for splenectomy are numerous. Splenectomies are performed in the treatment of polytrauma, tumor reduction surgery, or along with another abdominal surgery [2]. Splenectomies may be required for therapeutic purposes in the treatment of various hematological diseases [3], including constitutional hemoglobinopathies, auto-immune cytopenia [4,5,6], and lymphoproliferative [7] or myeloproliferative diseases [8]. Splenectomies can also be useful in rare situations, such as complicated infectious diseases (tuberculosis, malaria, Epstein Barr (EBV), or cytomegalovirus (CMV) infection), overload diseases [9,10], and cystic pathologies [11,12]. Recent literature reviews [3,8] have described these indications, which have evolved over the last decade [13].

The use of splenectomy as a diagnostic approach has been rarely described, as it is rarely performed [14,15,16,17]. Moreover, the major causes of splenomegaly of unknown origin appear to vary according to the study, as well as the geographic origin of the patients and the investigations performed [15,16,17]. Given the limited recent data in Europe concerning the use of diagnostic splenectomy, we analyzed the medical files of patients from two primary and tertiary care centers in France who underwent a diagnostic surgery and detail their characteristics and the diseases identified. The high frequency of lymphomas identified led us to suggest a diagnostic algorithm.

## 2. Materials and Methods

This study was conducted at Hopitaux Universitaires Paris–Saclay, AP-HP (CHU Bicêtre, 94270 Le Kremlin-Bicêtre, France) and the Percy Military Hospital (92140 Clamart, France). These two hospitals are both primary and tertiary care hospitals, housing internal medicine, clinical hematology, and visceral surgery departments. Medical data files from all patients who underwent splenectomy from January 2010 to December 2019 were retrospectively analyzed.

### 2.1. Inclusion and Exclusion Criteria

All consecutive patients aged 15 years or more who underwent a diagnostic splenectomy were included. Patients who had a therapeutic splenectomy (immune thrombocytopenic purpura (ITP), autoimmune hemolytic anemia (AIHA), lymphoproliferative syndromes, chronic myeloproliferative syndromes, spherocytosis, benign cysts) were excluded. Similarly, all patients who underwent a splenectomy following spleen trauma, a scheduled surgery with an operative spleen injury, or cytoreduction tumor surgery were excluded.

### 2.2. Outcomes

The primary endpoint of this study was the determination of the diagnostic ability of the splenectomy. Secondary endpoints were a description of the patients’ clinical, biological, and radiological characteristics and the pre-operative investigations and their diagnostic utility.

### 2.3. Variables and Analysis

The following variables were collected and analyzed for all patients: gender, age at the time of the splenectomy, date of symptom onset, initial care, time between the initial management and splenectomy, type of surgery (laparotomy or laparoscopy) and complications, presence of fever and/or B symptoms (defined by asthenia, fever, sweating, and weight loss), presence of splenomegaly and the size of the spleen (splenomegaly was defined either clinically or measured by morphological examinations (computed tomography or ultrasound) that confirmed a major axis > 12 cm), hepatomegaly, enlarged lymph nodes, and any other symptoms or clinical abnormalities. The results of biological analyses considered were those from the first medical contact. The collected biological characteristics were white blood cell, lymphocyte, eosinophil, monocyte, and platelet counts, hemoglobin, fibrinogen, ferritin, C-reactive protein (CRP), liver enzyme, calcemia, uric acid, and lactate dehydrogenase (LDH) levels, as well as plasma protein electrophoresis. The presence of hemophagocytic lymphohistiocytosis was sought, defined by the combination of clinical and biological signs [18]. Data from hematological investigations, such as lymphocyte phenotyping (peripheral blood or bone marrow), bone-marrow cytology, and biopsy, were also collected. Microbiological investigations were recorded: viral serology, blood cultures, and other microbial investigations. The results of radiological tests, including CT, FDG PET/CT, and trans-thoracic or trans-esophageal echocardiography, were also recorded. Other exams, such as pathology results of lymph nodes or bone-marrow biopsies, were recorded when performed. The date and status at the last follow-up were recorded.

### 2.4. Data Protection

The study was conducted according to the principles of the 1964 Declaration of Helsinki and its later amendments and fulfilled the French national ethics guidelines. The study was approved by the INDS/CNIL (ethics code: 2218207 v 0).

### 2.5. Statistical Analysis

Categorical and continuous data are presented as numbers (%) or medians (interquartile range, IQR), respectively. Intergroup comparisons were performed using Fisher’s exact test or the Mann–Whitney U test as appropriate, depending on the variable distribution. The statistical significance of the observations was set to *p* < 0.05. Data were analyzed and stored using GraphPad PRISM software (v.8.0, LaJolla, CA, USA).

## 3. Results

Fifty-five patients with splenectomy were identified at the Percy Military Hospital between 2010 and 2019, of which 46 were excluded because of a therapeutic indication. Eighty-seven patients were identified at the Bicêtre University Hospital, of which 76 splenectomies were excluded (Figure 1). In total, 122 therapeutic splenectomies were excluded, resulting in the 20 diagnostic splenectomies investigated in this study.

### 3.1. Demographics and Patient Characteristics

Thirteen men and seven women underwent diagnostic splenectomy via 12 laparotomies and eight laparoscopies. We considered an open approach with laparotomy if the surgery was converted into an open surgery after a laparoscopic approach.

Their characteristics are presented in Table 1. The median age at splenectomy was 65 (59.5; 76.5) years, with a median time of 5 (1.4; 14.3) months between the first medical consultation and the splenectomy. Significant medical histories consisted of low-grade non-Hodgkin lymphoma (NHL) (*n* = 1), diffuse large B cell lymphoma (DLBCL) (*n* = 1), and human immunodeficiency virus (HIV) (*n* = 1). Fever and B symptoms were present in 10/20 (50%) patients before the diagnostic splenectomy. The most common surgical indication was a combination of splenomegaly, constitutional symptoms (mainly fever), and cytopenia. Thirteen patients (65%) had unexplained splenomegaly. Ten (50%) had fever of unknow origin (FUO, defined according to the Petersdorf definition [19], lasting at least 3 weeks). The patients without splenomegaly (*n* = 7) were explored because of abdominal symptoms (*n* = 3), bicytopenia (*n* = 2), or suspicion of relapsing malignant hemopathies (DLBCL and low-grade NHL) (*n* = 2). Based on other clinical or radiological characteristics, hepatomegaly was present in 6/20 (30%) patients, peripheral adenopathies in 4/20 (20%), and deep adenopathies in 7/20 (35%). One patient had hemophagocytic lymphohistiocytosis.

### 3.2. Diagnostic Investigations Prior to Splenectomy

The diagnostic investigations performed and their efficiency, defined by the information provided to the investigation, are presented in Table 2. The 20 patients underwent a median of 10 (7–15) procedures before the splenectomy over a period of 5 (1.4; 14.3) months. Infectious investigations were carried out for 16 patients, consisting of serology to search for a number of infectious diseases: HIV (*n* = 14), hepatitis B virus infection (HBV) (*n* = 13), hepatitis C virus infection (HCV) (*n* = 13), EBV (*n* = 7), CMV (*n* = 8), toxoplasmosis (*n* = 3), B19 parvovirus (*n* = 1), borreliosis (*n* = 2), bartonellosis (*n* = 2), syphilis (*n* = 2), rickettsiosis (*n* = 1), and brucellosis (*n* = 2). We did not find any tests for Whipple’s disease. Three positive serologies were found: HIV 1 (*n* = 1), replicating hepatitis B (*n* = 1), and active hepatitis C (*n* = 1). Bacteriological examinations (blood cultures, cytobacteriological examination of urine or sputum) were performed for seven patients, of whom all but one had fever. All were negative. The search for mycobacteria was only mentioned in three cases and was found to be negative.

Pathological procedures included 28 biopsies and 13 bone-marrow aspirations. The most informative was bone-marrow biopsy, which was informative for four cases (suggesting lymphoma). Bone-marrow aspirations were consistently considered normal. The other biopsies performed were not informative, except for two. One, an axillary lymph-node biopsy, revealed a marginal zone lymphoma. The splenectomy was nevertheless performed due to hypermetabolic spleen nodules, suggestive on FDG PET/CT of a high-grade lymphoma. The other, consisting of a liver biopsy, revealed an abundant intra-sinusoidal T-lymphoid population. Splenectomy confirmed the diagnosis of T gamma-delta lymphoproliferative disease. Immunological investigations included phenotyping from 10 bone marrow or peripheral blood lymphocyte samples and were informative in six cases, showing monoclonal lymphocyte populations.

FDG PET/CT was the most frequently performed examination (*n* = 18) and revealed splenic hypermetabolism in 16 patients (89%). Among them, the splenectomy led to a definitive diagnosis for 15 (Table 2). Apart from splenic hypermetabolism, PET/CT identified other sites of FDG uptake for eight patients (tonsils, deep or superficial lymphadenopathy, discovertebral abnormalities, tissular hypermetabolisms) and guided surgical biopsies or punctures for six, but none of these targeted biopsies were informative.

Overall, among the 20 patients, the procedures performed before the splenectomy, apart from bone-marrow biopsy, helped to identify suspected non-Hodgkin’s B lymphomas (B-NHLs) in two patients after lymph-node or liver biopsy, but did not lead to any conclusions for the other 18 patients, thus leading to the diagnostic splenectomy.

### 3.3. Identified Pathologies

Splenectomy contributed to the final diagnosis in 19 of 20 cases. Lymphoid malignancies were the main pathology identified (Figure 2). Splenectomy mostly allowed the identification of B-NHL (*n* = 12/20), DLBCL (*n* = 8), and low-grade non-Hodgkin lymphoma (LG-NHL) (*n* = 4, two follicular lymphomas and two marginal zone lymphomas). Two patients had other lymphoid malignancies: one HL and one T gamma-delta splenic lymphoma. Two patients had a benign spleen tumor: a splenic hamartoma and a hemangioma. Splenectomy also revealed a histiocytic sarcoma, a pseudo-inflammatory tumor, and one nonspecific granulomatosis, classified in “other diseases”.

The splenectomy did not contribute to a diagnosis for only one of the 20 patients. This patient had splenomegaly, fever and B symptoms, and no other organ involvement. He was not pancytopenic, LDH levels were normal, and the FDG PET/CT showed FDG uptake of the spleen and other hypermetabolic sites, including the bones and infra-supradiaphragmatic lymph. This patient died of multi-organ failure without diagnosis a few days after surgery.

We compared patients with splenomegaly and those without (Table 3). Patients with splenomegaly more frequently had fever/B symptoms and multiple lymph nodes or hepatomegaly. Anemia (*n* = 12) and thrombocytopenia (*n* = 9) were significantly more frequent for patients with splenomegaly than for those without (anemia in four and thrombocytopenia in one). Leukopenia (*n* = 5), lymphopenia (*n* = 4), and pancytopenia (*n* = 4) were present in patients with splenomegaly but not in those without. Abnormal plasma protein electrophoresis (PPE) was observed for eight patients with splenomegaly, with one or more disorders, such as monoclonal dysglobulinemia, hypogammaglobulinemia, or an inflammatory profile. LDH levels were elevated in six patients. Patients without splenomegaly could have abnormal plasma protein electrophoresis (*n* = 3) but LDH levels were elevated for only one patient. Similarly, patients with splenomegaly more frequently had informative lymphocyte phenotyping. In most cases, PET/CT showed hypermetabolic spleens, regardless of the subgroup. We then compared the biological characteristics of the eight patients with DLBCL and the four with LG NHL (Table 4). There were no differences between the two groups in terms of cytopenia, inflammation, or splenic FDG uptake, but a trend towards higher LDH levels was observed for the DLBCL patients.

Five cases had postoperative complications—one paralytic ileus occlusion, one fatal ascitic fluid infection, two portal thromboses, and one hemoperitoneum. Five patients died in our cohort (all in the year following the splenectomy, except for one, who died four years later due to a relapse of DLBCL) but only the patient with the ascitic fluid infection died from an acute postoperative complication. Fourteen patients were still being followed-up in December 2019.

## 4. Discussion

This observational study focused on the investigation of the records of patients who underwent a diagnostic splenectomy between 2010 and 2019. Splenectomy was useful in 95% of cases, leading to a diagnosis, mainly B-cell malignancies.

Previous retrospective monocentric studies [14,15,16,17,20] have described cohorts of patients who underwent a diagnostic splenectomy. Among them, only two recent studies, one in the US and the other in India (2018/2019) presented the clinical and biological characteristics of the patients and their outcomes [15,16]. Interestingly, few patients underwent PET/CT, which has become a potent tool to investigate fever of unknown origin (FUO) [21].

In our study, the splenectomy allowed diagnosis in 19/20 (95%) cases, which is more than that reported in the previous studies, which arrived at a diagnosis in 70% to 80% of the cases: 79% in the recent Indian study, also performed over a 10-year period [16]. The percentage of diagnoses found in our study was higher than that in the American study published in 2019 [15], which found a rate of diagnosis of only 46%, perhaps because the prior diagnostic workup was different. In common with this study, the main diagnosis after surgery was B-cell lymphoma.

We observed mostly hematological malignancies (14 of 20 cases) and the absence of infectious diseases. These results are different from those of previous studies conducted in India [16] and in China [14,20], which found infectious causes, particularly mycobacterial infections (tuberculosis or others). This discrepancy may be explained by the difference between the prevalence of chronic infections in France and that of tropical or sub-tropical countries.

In the setting of patients with difficult diagnoses, such as FUO or unexplained constitutional symptoms (*n* = 10/20 patients in our series), the infectious workup is very important and must include serology studies (at least HIV, HBV, and HCV), the search for mycobacterial or tropical diseases, depending on the history of the patients, and blood cultures or other bacteriological exams, depending on the clinical examination (Figure 3). In the clinical presentation of the patients reported here, the two major clinical parameters were splenomegaly and B signs, similar to other reports [14,15]. Serology was performed for most patients in our series (16/20) and found HIV infection in one patient and chronic hepatitis B/C in two. The search for bacteria or mycobacteria was scarcely reported (*n* = 7 patients with fever) in our centers and were negative. This observation is different from those of previous studies, such as that conducted in India (16), which led to 13 diagnoses of infectious disease (tuberculosis, melioidosis, candidal abscess).

However, second- or third-line workup to search for infectious disease, including serology for various rare diseases (bartonellosis, borreliosis, brucellosis) may be useful in such a setting of FUO or unexplained splenomegaly. In case of splenomegaly without FUO, lysosomal storage diseases have to be searched especially if there is a familial history.

In parallel with microbial investigations, FDG PET/CT has become an important diagnostic tool. Indeed, in the setting of FUO/inflammation of unknown origin (IUO), the diagnostic workup has changed markedly over the last 40 years. Although bone-marrow or liver biopsies were considered to be useful in the 2nd or 3rd line in the 1980s and 1990s, the use of FDG PET/CT has increasingly grown since the mid-2000s [22]. Here, we highlight the important place of PET/CT, which led to diagnostic information in 89% of the cases. Particularly in cases of FUO, PET/CT has provided important diagnostic information [23], increasing the final rate of diagnosis to a very high proportion, depending on the study, from more than 50% [24] to 77.4% [25]. FDG PET/CT imaging may be useful in guiding biopsies and optimizing the diagnostic workup.

Bone-marrow biopsy showed high diagnostic efficacy and should be performed if a hematological disease is suspected [26]. It is still an informative examination, especially in countries where PET/CT is not easily available. However an Australian study [27] reported that performing a bone-marrow biopsy with no other hematological abnormalities in the context of FUO is of very limited value and they suggested first using PET/CT to help guide the diagnostic process.

The frequency of hematological malignancies we found also underscores the option of performing detailed flow cytometry analysis of the blood and bone marrow, allowing the detection of abnormal cells and orienting the diagnostic workup.

Thus, we propose standard biological and radiological investigations and an algorithm to guide the compulsory investigations and indicate when to perform a diagnostic splenectomy (Figure 3) for patients without a current diagnosis and with clinical symptoms (fever, B symptoms, splenomegaly, abdominal symptoms) lasting more than three weeks, such as the definition of FUO.

Based on this algorithm, splenectomy helped the clinicians to arrive at the diagnosis for 19 patients, including 16 with spleen FDG uptake on PET/CT. The diagnosis of low-grade B-NHL was already suspected for two patients, but the splenectomy was performed to disclose Richter’s syndrome.

Aside from splenectomy, spleen biopsies could also be considered in cases of focal spleen lesions on PET/CT. Recent studies [28,29] reported excellent sensitivity and specificity without major complications. Splenic biopsy is, nonetheless, a procedure with potential complications, such as hemorrhage, and it should be carried out by an experienced radiologist.

Concerning the post-operative outcomes after splenectomy, five patients had early complications. The previously described studies reported equivalent rates of post-operative complications: 25.9% [14] and 26.5% [20]. These results highlight that splenectomy is still a risky procedure and that it should only be performed once other investigations have proven to be unsuccessful.

Thromboembolic complications are common following splenectomy and we noted two portal thromboses in our series, as previously reported [30]. To reduce the rate of such complications, patients should receive an extended thromboprophylaxis course.

Our study had several limitations. The retrospective design and small size of our cohort limits the strength of the results, as some data concerning the number and types of certain procedures performed may be missing. In addition, the etiological investigations prior to splenectomy were not standardized. Our study is illustrative of Western European clinical practice and the conclusions cannot be extended to other geographical areas. The difference between our results and those of a recent publication from India, in which the authors reported a high frequency of mycobacterial infections, is striking in this setting [16].

## 5. Conclusions

In conclusion, splenectomy may be useful for the diagnosis of FUO and unexplained splenomegaly, but it should be performed only after exhaustive investigations. In our study, the splenectomy contributed to the diagnosis mainly of lymphoproliferative disorders, especially diffuse large B-cell lymphomas, which can be limited to the spleen. Our analysis of the pre-operative investigations underscores the importance of the pre-operative work-up. Our study highlights the usefulness of FDG PET/CT in these situations and the need for complete hematological investigations and proposes an algorithm that can be used before performing a diagnostic splenectomy.

## Figures and Tables

**Figure 1 jcm-10-01519-f001:**
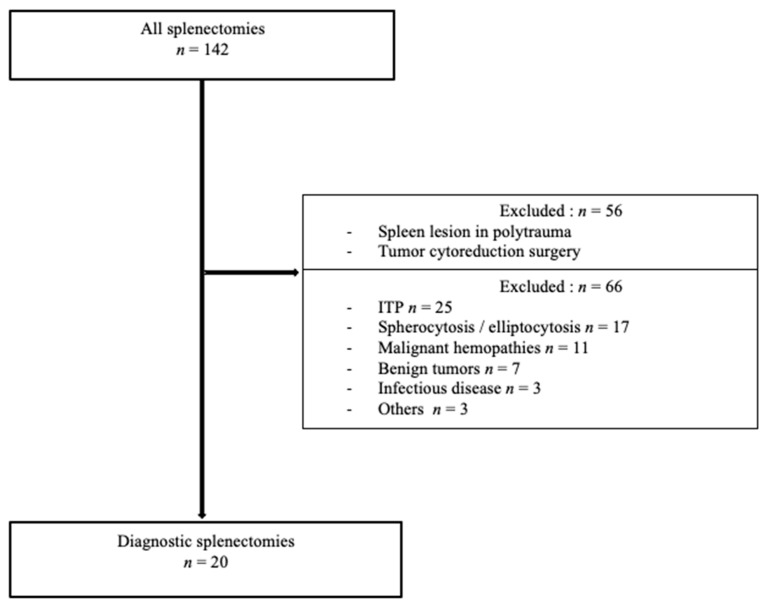
Flow chart. ITP, immune thrombocytopenic purpura.

**Figure 2 jcm-10-01519-f002:**
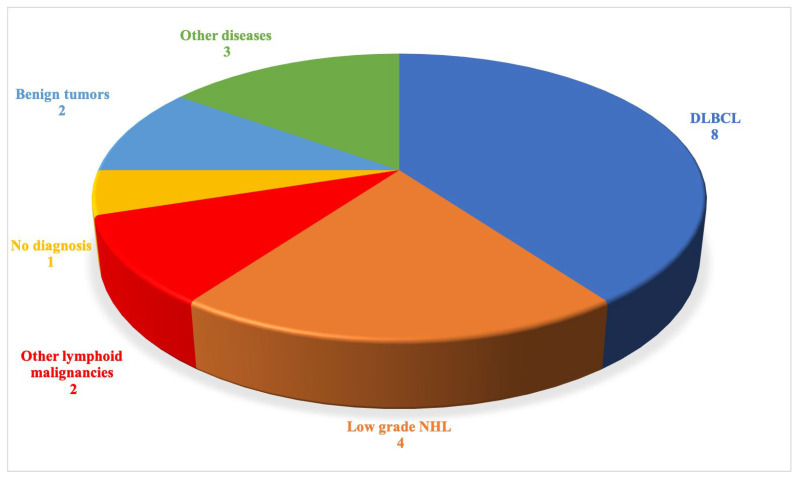
Identified pathologies (represented as the absolute number of patients).

**Figure 3 jcm-10-01519-f003:**
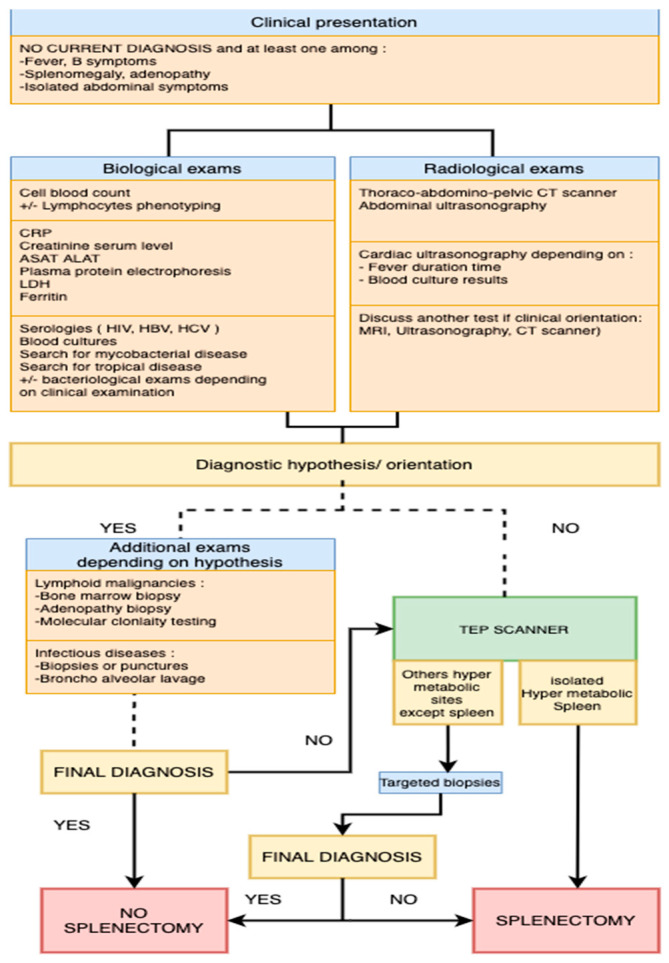
Diagnostic splenectomy: an algorithm to help clinical decision-making according to first-line examinations.

**Table 1 jcm-10-01519-t001:** Patient demographic, clinical, and biological characteristics.

Male/Female, *n* (%)	13/7 (65/35)
Laparoscopy, *n* (%)	8 (40)
Laparotomy, *n* (%)	12 (60)
Median age at splenectomy (years) Median (interquartile range (IQR))	65 (59.5–76.5)
Splenomegaly, *n* (%)	13 (65)
Hepatomegaly, *n* (%)	6 (30)
Enlarged lymph nodes, *n* (%)	7 (35)
Fever/B symptoms, *n* (%)	10 (50)
Other symptoms, *n* (%) *	4 (20)
Leukopenia (leucocytes < 4 × 10^9^/L), *n* (%)	5 (26)
Lymphopenia (lymphocytes < 1.5 × 10^9^/L), *n* (%)	4 (24)
Anemia (hemoglobin < 12 g/dL), *n* (%)	16 (84)
Thrombopenia (<150 × 10^9^/L), *n* (%)	10 (53)
Abnormal plasma protein electrophoresis	11 (79)
Elevated level of LDH (>430 IU/L), *n* (%)	7 (41)

* Other organ involvement: skin edema (*n* = 1), pleural effusion (*n* = 1), cardiac dysfunction and hemorrhagic cutaneous syndrome (bruising) (*n* = 1), and ascites (*n* = 1). IQR, interquartile range. LDH, lactate dehydrogenase.

**Table 2 jcm-10-01519-t002:** Diagnostic investigations prior to the splenectomy and their results concerning the 20 patients.

Diagnostic Procedure for Pathology Prior to Splenectomy	N. of Performed Procedures	N. of Informative Procedures	Suspected Diagnosis after Procedure or Contributive Information	Final Diagnosis after Splenectomy
Bone-marrow biopsy	13	4	3 suggestive of LG NHL	2 DLBCL complicating LG NHL, 1 LG NHL
			1 suggestive of T-cell lymphoma	1 pseudo-inflammatory tumor
Liver biopsy	2	1	1 suggestive of T-cell lymphoproliferation	1 T gamma-delta splenic lymphoma
Superficial lymph-node biopsies	3	1	1 suggestive of marginal zone lymphoma	1 marginal zone lymphoma
Lymphocyte phenotyping	10	6	4 suggestive of LG NHL,	3 LG NHL, 1 DBCL complicating LG NHL
			1 suggestive of secondary LGL	1 histiocytic sarcoma
			1 suggestive of T-cell lymphoproliferation	1 T gamma-delta splenic lymphoma
Microbial investigations	16	3	1 HIV-1	
			1 replicating hepatitis B	No infectious disease identified
			1 active hepatitis C	
FDG PET/CT	18	16	16 hypermetabolic spleens, 8 patients with other hypermetabolic loci (tonsils, deep or superficial lymphadenopathy, vertebral disco fixation, muscular tissue fixation)	6 DLBCL 4 LG NHL 1 Hodgkin lymphoma 1 T gamma delta splenic lymphoma 1 pseudo-inflammatory tumor 1 hemangioma 1 granulomatosis 1 without diagnosis
Echocardiography	7	0	None	Not applicable
Bone-marrow aspiration	10	0	None	Not applicable
Karyotype	5	0	None	Not applicable
Other sites biopsies (Bones, gastric and duodenal, spleen, temporal artery)	10	0	None	Not applicable

LG: low grade, HL: Hodgkin lymphoma, NHL: non-Hodgkin lymphoma, LGL: large granular lymphocytes, DLBCL: diffuse large B-cell lymphoma. FDG PET/CT, ^18^F-fluorodeoxyglucose positron emission tomography/computed tomography.

**Table 3 jcm-10-01519-t003:** Patient clinical and biological characteristics, investigations, and diagnoses according to clinical presentation *.

	Patients with Splenomegaly *n* = 13	Patients without Splenomegaly *n* = 7	*p*
Clinical presentation			
Fever/B symptoms, *n* (%)	8 (62)	2 (29)	0.35
Hepatomegaly, *n* (%)	6 (46)	0 (0)	0.052
Lymph nodes, *n* (%)	9 (69)	2 (29)	0.16
Other organ involvement, *n* (%)	3 (23)	1 (14)	1
Isolated abdominal symptoms, *n* (%)	0 (0)	3 (43)	0.03
Biological presentation			
Anemia, *n* (%)	12/12 (100)	4/7 (57)	0.04
Thrombopenia, *n* (%)	9/12 (75)	1/7 (14)	0.02
Leukopenia, *n* (%)	5/12 (42)	0/7 (0)	0.11
Lymphopenia, *n* (%)	4/10 (40)	0/7 (0)	0.10
Pancytopenia, *n* (%)	4/12 (33)	0/7 (0)	0.25
Abnormal plasma protein electrophoresis, *n* (%)	8/10 (80)	3/4 (75)	1
Elevated level of LDH, *n* (%)	6/12 (50)	1/5 (20)	0.34
Positive pre-surgery investigations			
Median (IQR) duration of investigations (months)	5 (2.25–29.5)	5 (1–14)	0.55
Lymphocyte phenotyping, *n* (%)	5/8 (63)	1/2 (50)	1
PET/CT with hypermetabolic spleen, *n* (%)	10/12 (83)	6/6 (100)	0.53
Bone-marrow biopsy, *n* (%)	3/10 (30)	1/3 (33)	1
Identified diseases			
Final diagnosis, *n* (%)	12 (93)	7 (100)	1
Lymphoma, *n* (%)	10 (77)	4 (57)	0.61
DLBCL, *n* (%)	6 (46)	2 (29)	0.64
Low-grade NHL, *n* (%)	3 (23)	1 (14)	0.60

Anemia was defined as Hb < 12 g/dL for women/<13 g/dL for men; leukopenia was defined as a WBC < 4 × 10^9^/L; lymphopenia was defined as a lymphocytes count < 1.5 × 10^9^/L; thrombocytopenia was defined as a platelet count < 150× 10^9^/L; an elevated level of LDH was defined as an LDH level > 430 IU/L), * data were not available for all the patients.

**Table 4 jcm-10-01519-t004:** Clinical and biological characteristics of patients with DLBCL compared to those with low-grade NHL diagnosed after splenectomy *.

	Patients with DLBCL (*n* = 8)	Patients with Low-Grade NHL (*n* = 4)	*p*
Median (IQR) duration between first medical contact and surgery (months)	2.25 (1–12.75)	3.5 (2.5–5.25)	0.93
B signs/fever, *n* (%)	3 (38)	2 (50)	1
Splenomegaly, *n* (%)	6 (75)	3 (75)	1
Leukopenia, *n* (%)	3/7 (43)	1 (25)	1
Anemia, *n* (%)	6/7 (86)	3 (75)	1
Thrombocytopenia, *n* (%)	5/7 (71)	2 (50)	0.58
Elevated LDH levels, *n* (%)	6 (75)	0 (0)	0.06
Elevated C-reactive protein, *n* (%)	3/4 (75)	1 (25)	0.49
Splenic PET-hyperfixation, *n* (%)	6/7 (86)	2 (50)	0.49

Anemia was defined as Hb < 12 g/dL for women/<13 g/dL for men; leukopenia was defined as a WBC < 4 × 10^9^/L; thrombocytopenia was defined as a platelet count < 150 × 10^9^/L; an elevated level of LDH was defined as an LDH level > 430 IU/L). Elevated C-reactive protein was defined as a CRP level > 5 mg/L; * data were not available for all the patients.

## Data Availability

The data presented in this study are available on request from the corresponding author.

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
