# Peer review of "Diagnostic Splenectomy: Characteristics, Pre-Operative Investigations, and Identified Pathologies for 20 Patients"

_jcm, 2021, doi:10.3390/jcm10071519_

Round 1
Reviewer 1 Report
The authors report a series of 20 diagnostic splenectomies during 10 yrs in two french hospitals.
This is a short series but of interest due to the unfrequent clinical situation, in centers when splenectomy is the last resource after a complete clinical and hemathological workup.
main complain.
- There are NO medical or surgical splenectomies. All the splenectomies are SURGICAL. Ther are diseases in which, to complete the diagnosis merit a surgical procedure to obtain it.
- Whta means splenomegaly, how it was definede (palpation, CT scan, measures, postop weight of the spleen), and which kind of enlarged spleen (massive, supra massive), which was the pathological measures (weigth and size)
- Currently slenectomy is routinely performed by laparoscopy. This detail matters because the outcome is better after lap surgery, and sometimes facilitates its indication. Why 60% of the patiente were oeprated trough an open approach??
- The results and discussion is trough wordy, and it may be resumed in tables (results section), or shortened (discussion)
Author Response
Review report 1 :
- There are NO medical or surgical splenectomies. All the splenectomies are SURGICAL. There are diseases in which, to complete the diagnosis merit a surgical procedure to obtain it.
Response: We agree with the reviewer. It was actually confusing. We changed the “medical and surgical” terms by “diagnostic” and “therapeutic approaches”. We also modified the flow chart accordingly.
- What means splenomegaly, how it was defined (palpation, CT scan, measures, postop weight of the spleen), and which kind of enlarged spleen (massive, supra massive), which was the pathological measures (weight and size)
Response: We defined these terms in the section “2.3 Variables and analysis” in page 2 Line 83: « splenomegaly was defined either clinically or measured by morphological examinations (computed tomography or ultrasound) that confirmed a major axis > 12 cm »
- Currently splenectomy is routinely performed by laparoscopy. This detail matters because the outcome is better after lap surgery, and sometimes facilitates its indication. Why 60% of the patients were operated through an open approach?
Response: We fully agree with the reviewer and thank him/her for this helpful comment. Some patients had massive splenomegalies and the open surgery was first decided. We also considered an open approach if the surgery was converted into an open surgery after a laparoscopic approach. We mentioned it in the manuscript, page 3 line 119: “We considered an open approach with laparotomy if the surgery was converted into an open surgery after a laparoscopic approach”.
- The results and discussion is trough wordy, and it may be resumed in tables (results section), or shortened (discussion)
Response : We have shortened the discussion accordingly. The following sentences have been removed:
Line 236: « In terms of the pre-operative work-up, these two studies described the number of patients who underwent various examinations (PET/CT, bone-marrow biopsy, and lymph-node biopsy) but the results were not reported in detail (certain examinations, such as serology and mycobacterial investigations were not reported). »
Line 247/248 : between 2017 and 2019 and between 2006 and 2015 and 1982 and 2005
Line 262 : This discrepancy may be explained by the difference between the prevalence of chronic infections in France and that of tropical or sub-tropical countries. It may also be related to our definition of FUO: most of the patients had negative results from first-line microbial investigations in ambulatory care or other primary care centers before being referred to our hospitals.
Line 276 : The recent studies led in the USA (15) and India (16) reported only limited use of PET/CT for 10/35 and 2/38 patients, respectively.
Line 282 : FDG PET/CT appears to be a better first-line examination modality than certain invasive procedures, such as liver biopsies, which were, however, largely performed in the 80′s (28,29).
Line 285 : Access to such analysis modalities is better today than ten years ago and they are helpful.
Line 304/306 : Several studies confirmed the need to target focal lesions to make the biopsy useful (30,32). In our cohort, the patients had diffuse splenomegaly but rarely a focal lesion.
Line 307 : the expected diagnostic utility must be balanced with the potential risks. Moreover,
Line 319/321 : The patients in our cohort were all vaccinated and were treated by antibiotic prophylaxis. As in the studies cited above (15,16), we did not find encapsulated germ infections or late sepsis. Nonetheless, our follow-up period was short and we know that late infection can occur, up to 12 years after surgery (33).
Reviewer 2 Report
In this paper, the authors describe the use of splenectomy as a diagnostic tool in cases of splenomegaly and fever in which it was not possible to make a certain diagnosis with the available diagnostic techniques. The authors present very well all the diagnostic procedures performed before the splenectomy. The fact that splenectomy allowed diagnosis in 19/20 patients is very tempting.
In any case, splenectomy is an important surgery which, especially on "fragile" patients, can cause serious complications. Furthermore, as pointed out by the authors, it has very serious long-term infectious and thrombotic complications. In this regard, the authors describe 5 patients with early complications (one of which is fatal) and 4 patients who died within 1 year of surgery:"Five cases had postoperative complications: one paralytic ileus occlusion, one fatal ascitic fluid infection, two portal thromboses, and one hemoperitoneum. Five patients died (all in the year following the splenectomy, except for one, who died four years later. Fourteen patients were still being followed-up in December 2019". Is there a relationship between these groups or only one patient died for postoperative complications? It should be better specified.
It would be worth mentioning the recent molecular diagnostic techniques (i.e. NGS, exome sequencing). To date, new molecular markers of the disease are identified continuously. They allow to made a precise and accurate diagnosis. Were they performed on any of the cases in which the diagnostic splenectomy was done? if so, what were the results? It would be better to describe them
Author Response
Review report 2 :
- In any case, splenectomy is an important surgery which, especially on "fragile" patients, can cause serious complications. Furthermore, as pointed out by the authors, it has very serious long-term infectious and thrombotic complications. In this regard, the authors describe 5 patients with early complications (one of which is fatal) and 4 patients who died within 1 year of surgery: «Five cases had postoperative complications: one paralytic ileus occlusion, one fatal ascitic fluid infection, two portal thromboses, and one hemoperitoneum. Five patients died (all in the year following the splenectomy, except for one, who died four years later. Fourteen patients were still being followed-up in December 2019". Is there a relationship between these groups or only one patient died for postoperative complications? It should be better specified.
Response: We thank the reviewer, this is indeed an important point to clarify. Only the patient with the ascitic fluid infection died from the postoperative infection. We clarify in the manuscript. « Five patients died in our cohort (all in the year following the splenectomy, except for one, who died four years later due to a relapse of DLBCL) but only the patient with the ascitic fluid infection died from the acute postoperative complication » page 8, line 225
- It would be worth mentioning the recent molecular diagnostic techniques (i.e. NGS, exome sequencing). To date, new molecular markers of the disease are identified continuously. They allow to made a precise and accurate diagnosis. Were they performed on any of the cases in which the diagnostic splenectomy was done? if so, what were the results? It would be better to describe them
Response :This is an interesting question. We did not find any recent molecular diagnostic technique like NGS or exome sequencing in the files of our patients, but their diagnostic work up spans between years 2010 and 2019. These recent techniques should certainly be added in the current approach. We add the molecular clonality testing in the diagnostic algorithm (Figure 3).
Reviewer 3 Report
- Brief summary
This article aimed at identifying the diagnostic ability of splenectomy in patients with a non-diagnostic work-up, as the efficacy of splenectomy when used as a diagnostic test is not well-established. The results obtained are very interesting as lymphomas were more frequently encountered, thus helping the clinicians in their reasoning for undiagnosed patients. In addition, the article is supplemented by a useful clinical algorithm before leading to splenectomy.
In this context, some major critical comments and suggestions for changes are necessary. - Major criticism
It is strongly recommend that the authors clarify the terms "surgical" and "medical" splenectomy, as both can be performed for therapeutic purposes as indicated in the introduction. These are certainly regional medical terms, but this differentiation has no scientific equivalent and makes the understanding more confusing. The splenectomy realized in a context of a diagnosis could also been therapeutic, for instance in a SMZL to avoid complications such as auto immune anemia. In addition, readers may ask why they have been differentiated while only diagnostic splenectomies were studied.
The authors should carefully read the instructions concerning table and figure captions : “They should provide a description of the object such that the reader does not need to refer to the main text to fully understand”. Please elaborate the captions for tables and figures.
The English should be carefully reviewed as some literal translations from French were observed.
The authors sometimes compared percentages to other studies, sometimes ordinal variables (e.g. number of patients) particularly in the fifth paragraph of the discussion. Please compare similar data and specify whether or not the studies are comparable, as inclusion criteria often vary across them.
3. Minor points of criticism
Line 42 : As mentioned in the major criticism part, please clarify the “surgical” and “medical” terms.
Line 67: The “therapeutic” term is used therefore this term should appear in figure 1 instead of medical splectomies.
Line 74 : The diagnostic “utility”: did the authors mean “diagnostic ability”?
Line 79 : “date of first symptoms”. “Symptom onset date ? “
Line 125 : general symptoms: Please clarify.
Table 1 :
- Mixing Patient demographic and biological characteristics does net help the reader understanding the key message. Taken in isolation, the biological characteristics are difficult to interpret given the small sample of patients, as many etiologies could explain the biological alterations (LDH, anemia, ELP serum abnormalities). (LDH, Anemia, serum protein electrophoresis,…). Interestingly, the percentage of diagnosis found after splenectomy (95%) is a valuable information but is situated last in the table. However, this information should not appear in this table since it does not correspond to the table heading.
- Caption is written twice. Please provide a more detailed description. Please read instructions for authors.
Table 2 :
- Captions : Precise on which population this table is (the 20 patients? ).
- Develop the acronym HIV.
- However, the diagnostic "biopsies" procedures below the PET/CT category should be grouped/reduced into a single category since these different biopsy sites are patient-dependent, uninformative and make it more complex to understand the table.
Figure 2: the caption had been included in the figure itself. Please modify.
Table 3.
- Please precise “characteristics”. Subcategories should be identified more clearly.
- Please use the international standard units ( mm3 ==> 109/L)
- LDH: Describe the upper value studied below the table as the other parameters.
Table 4:
- Please precise “characteristics” : biological ?
- “ Inflammation” : Please precise : C-reactive protein ?
- The total number of patients in the column “patients with DLBCL” does not always correspond to the same number of patients for the criterion studied. The percentage given is therefore not representative of the population studied and misleads the reader. Please modify/clarify.
Line 233-238 : Precise if the results obtained and those from the two studies were strictly comparable or not. How would the authors explain the difference of percentage of diagnosis with the American study as they observe also the same main diagnosis? Hypothesis are needed for such a difference.
Line 244 : “FUO origin” . “Origin” should be deleted.
Line 245 : “altered general status”. This is not scientifically considered as a symptom. Please clarify or provide a scoring system of performance status.
Line 256 : the “frequency” : did you mean prevalence ?
Figure 3.
- Please develop the caption.
- A left arrow is probably missing above the second box on the left?
- Is there any data available about the fact that PET/CT-scan decreases the rate of diagnostic splenectomy? This graph highlights that a PET/CT-scan showing an isolated hypermetabolic spleen in undiagnosed patients is an indication for splenectomy. Would you strongly recommend this pathway taking into account the possible postoperative complications ?
Line 266: IUO: please develop the acronym as it is first encountered.
Were there any other benefits to splenectomy other than getting the diagnosis in these 20 patients? It would be interesting to mention this since diagnostic splenectomies can be partly therapeutic.
- Based on your observations of the 19 patients who underwent diagnostic splenectomy, is the pre-test probability of lymphoid hemopathy therefore higher in undiagnosed patients with splenomegaly? Please elaborate.
Author Response
Review report 3 :
- It is strongly recommend that the authors clarify the terms "surgical" and "medical" splenectomy, as both can be performed for therapeutic purposes as indicated in the introduction. These are certainly regional medical terms, but this differentiation has no scientific equivalent and makes the understanding more confusing. The splenectomy realized in a context of a diagnosis could also been therapeutic, for instance in a SMZL to avoid complications such as auto immune anemia. In addition, readers may ask why they have been differentiated while only diagnostic splenectomies were studied.
Response: We agree with the reviewer. It was actually confusing. We changed the “medical and surgical” terms by “diagnostic” and “therapeutic” approaches. We also modified the flow chart accordingly.
- The authors should carefully read the instructions concerning table and figure captions : “They should provide a description of the object such that the reader does not need to refer to the main text to fully understand”. Please elaborate the captions for tables and figures.
Response: We thank the reviewer for this comment and we clarified the captions for tables and figures
- The English should be carefully reviewed as some literal translations from French were observed.
Response : we changed the literal translations and we tried to improve the whole manuscript
- -The authors sometimes compared percentages to other studies, sometimes ordinal variables (e.g. number of patients) particularly in the fifth paragraph of the discussion. Please compare similar data and specify whether or not the studies are comparable, as inclusion criteria often vary across them.
-Line 233-238 : Precise if the results obtained and those from the two studies were strictly comparable or not. How would the authors explain the difference of percentage of diagnosis with the American study as they observe also the same main diagnosis? Hypothesis are needed for such a difference.
Response : We agree with the reviewer, adding this point and compare and detail the studies can be useful but we have to shorten the discussion so it’s not easy to develop but we suggested the diagnostic workup was different adding « perhaps because the prior diagnostic workup was different ».
- Line 42 : As mentioned in the major criticism part, please clarify the “surgical” and “medical” terms.
Line 67: The “therapeutic” term is used therefore this term should appear in figure 1 instead of medical splenectomies.
Response : As pointed above, we have modified the text accordingly.
- Line 74 : The diagnostic “utility” : did the authors mean “diagnostic ability” ?
Response: we mean diagnostic ability and we changed it in the manuscript
- Line 79 : “date of first symptoms”. “Symptom onset date ? “
Response: We thank the reviewer for this clarification and we changed it by “ date of symptoms onset”
- Line 125 : general symptoms : Please clarify.
Response: We mean « constitutional symptoms » like fever, weakness and weight loss. We changed this term.
- Table 1 :
- Mixing Patient demographic and biological characteristics does net help the reader understanding the key message. Taken in isolation, the biological characteristics are difficult to interpret given the small sample of patients, as many etiologies could explain the biological alterations (LDH, anemia, ELP serum abnormalities). (LDH, Anemia, serum protein electrophoresis,…). Interestingly, the percentage of diagnosis found after splenectomy (95%) is a valuable information but is situated last in the table. However, this information should not appear in this table since it does not correspond to the table heading.
- Caption is written twice. Please provide a more detailed description. Please read instructions for authors.
Response: We removed the percentage of diagnosis found after splenectomy in Table 1 but we think it is important to keep biologic results. As the reviewer pointed, these data are not discriminating so this underlies the importance to perform a large set of investigations and ultimately the splenectomy. We removed the caption in the table and precised the caption. The title of Table 2 was changed: “Table 2. Diagnostic investigations prior the splenectomy and their results concerning the 20 patients”
- Table 2 :
- Captions : Precise on which population this table is (the 20 patients? ).
Response: The population is all the cohort so the 20 patients. We specified it
- Develop the acronym HIV.
Response: We developed this acronym, as well as the others not developed acronyms (such as HBV, HCV or EBV).
- However, the diagnostic "biopsies" procedures below the PET/CT category should be grouped/reduced into a single category since these different biopsy sites are patient-dependent, uninformative and make it more complex to understand the table.
Response: We agree with the reviewer and simplified the message so we reduced into a single category “ Other sites biopsies”
- Figure 2: the caption had been included in the figure itself. Please modify.
Response: we removed the caption included in the figure
- Table 3.
- Please precise “characteristics”. Subcategories should be identified more clearly.
Response: We changed the caption « Patient clinical and biological characteristics, investigations, and diagnoses according to clinical presentation. »
- Please use the international standard units ( mm3 ==> 109/L). Response: We modified accordingly the international standard units 109/L
- LDH: Describe the upper value studied below the table as the other parameters.
Response: We precised that Elevated level of LDH was defined as a LDH level > 430 IU/l
- Table 4:
- Please precise “characteristics” : biological ?
- “ Inflammation” : Please precise : C-reactive protein ?
- The total number of patients in the column “patients with DLBCL” does not always correspond to the same number of patients for the criterion studied. The percentage given is therefore not representative of the population studied and misleads the reader. Please modify/clarify. Response: we changed the caption by “ Clinical and biological characteristics of patients with DLBCL compared to those with low-grade NHL diagnosed after splenectomy.”
We removed inflammation and replaced it by C-reactive protein
In the table 3 and 4, all data were not available so we filled in the table with the number of patients we had the data. To clarify, we added an * in the title of these tables and added in the footnotes, the sentence : * data were not available for all the patients.
- Line 244: “FUO origin” . “Origin” should be deleted.
Response: we deleted the word and let the acronym
- Line 245: “altered general status”. This is not scientifically considered as a symptom. Please clarify or provide a scoring system of performance status.
Response: We mean « constitutional symptoms » like fever, weakness and weight loss. We changed this term
- Line 256: the “frequency” : did you mean prevalence ?
Response: we indeed meant prevalence and we changed the term.
- Figure 3.
- Please develop the caption.
- A left arrow is probably missing above the second box on the left?
- Is there any data available about the fact that PET/CT-scan decreases the rate of diagnostic splenectomy? This graph highlights that a PET/CT-scan showing an isolated hypermetabolic spleen in undiagnosed patients is an indication for splenectomy. Would you strongly recommend this pathway taking into account the possible postoperative complications ?
Response: We developed the caption, removed the arrow on the second box on the right.
Also, we thank the reviewer for this interesting comment. There is no published data to confirm the PET/CT decreases the rate of the diagnostic splenectomy. In our study the PET/CT helps to choose between targeting another site for biopsy or confirming the only metabolism on the spleen. For patients with an urgent need of diagnosis and an isolated hypermetabolism in the spleen at the PET/CT, we would recommend to perform the splenectomy. The high frequency of NHL found in our study supports such a suggestion.
- Line 266: IUO: please develop the acronym as it is first encountered.
Response: we developed the acronym : « Inflammation of unknown origin »
- Were there any other benefits to splenectomy other than getting the diagnosis in these 20 patients? It would be interesting to mention this since diagnostic splenectomies can be partly therapeutic.
Response: We agree with the reviewer but the first benefit is to diagnose in order to treat the patients. No patient in our study had a limited disease and was cured only by the surgery but it could be possible, especially in the setting of splenic lymphomas for example.
- Based on your observations of the 19 patients who underwent diagnostic splenectomy, is the pre-test probability of lymphoid hemopathy therefore higher in undiagnosed patients with splenomegaly? Please elaborate.
Response: This is an interesting question. However, it seems difficult to give a robust answer because of the small number of patients studied.
Reviewer 4 Report
This is a well written and presented work about diagnostic splenectomy. This topic is rarely systematically analyzed. Of special interest is the fact, that novel diagnostic tools like PET-CT and flowcytometry are integrated parts of the diagnostic work up. This might be the reason for the high number of specific diagnosis gained by the procedure. As the authors mention in their paper, geographic differencies are important and therefore the conclusions are more or less mainly valid for non-tropical, western countries. I recommend to change the manuscript on three minor points:
- Because the diagnosis was mainly B-NHL, I would recommend to do in all patients with no easy accesible lymphnode a bone marrow with biopsy and flow cytometry within the bone marrow aspirate. Of additional help is a molecular genetic test for clonality (T and B lymphocytes). Small T-clones can be of great importance in cases with T-LGL disease.
- in patients with just splenomegaly (and often thrombocytopenia) but no FUO, lysosomal storage disease like Gaucher should be excluded.
- the authors should adress the important point of how to reduce secondary thrombotic complications after splenectomy.
Since the 2016 WHO classification includes several entities within the B-NHL, which are only diagnosable by splenectomy, the procedure will remain an important diagnostic procedure.
Author Response
Review report 4 :
- Because the diagnosis was mainly B-NHL, I would recommend to do in all patients with no easy accessible lymph node a bone marrow with biopsy and flow cytometry within the bone marrow aspirate. Of additional help is a molecular genetic test for clonality (T and B lymphocytes). Small T-clones can be of great importance in cases with T-LGL disease.
Response: We fully agree with the reviewer and thank him/her for this helpful comment. We added the molecular clonality testing in the diagnostic algorithm. The bone marrow and Flow cytometry are indeed, our second line of examinations after the first line of biological and radiological tests.
- In patients with just splenomegaly (and often thrombocytopenia) but no FUO, lysosomal storage disease like Gaucher should be excluded.
Response: we added this point in the manuscript. “In case of splenomegaly without FUO, lysosomal storage diseases have to be searched especially if familial history. » page 8 Line 265
- The authors should address the important point of how to reduce secondary thrombotic complications after splenectomy.
Response: We agree, this is an important message. We added : “To reduce the rate of that complication patients should receive an extended thromboprophylaxis course » page 11 line 313
Round 2
Reviewer 1 Report
Accepted with the corrections included by the authors
Author Response
Response : We thank the reviewer for the valuable comments which helped us to improve the manuscript